

# Comparative analysis of fecal microbiota of central and eastern black-necked cranes (*Grus nigricollis*) wintering in Yunnan Province, China

Ruimei Wang[1,2,*], Yixuan Wang[2,*], Lulu Deng[1], Binghui Wang[3], Mingfei Shi[1,2], Zeya Yang[1,2], Dong Hu[4], Zijiao Zhao[5], Ruiling Yuan[2] and Jiuxuan Zhou[2]

[1] Faculty of Life Science and Technology, Kunming University of Science and Technology, Yunnan, Kunming, China
[2] Yunnan Academy of Forestry and Grassland, Yunnan, Kunming, China
[3] School of Public Health, Kunming Medical University, Yunnan, Kunming, China
[4] Forestry and Grassland Bureau of Zhaotong City, Yunnan, Zhaotong, China
[5] Dashanbao National Nature Reserve Management and Protection Bureau, Yunnan, Zhaotong, China
* These authors contributed equally to this work.

Corresponding authors
Ruiling Yuan,
yuanruiling@yafg.ac.cn
Jiuxuan Zhou,
zhoujiuxuan1@yafg.ac.cn

## ABSTRACT

The black-necked crane (*Grus nigricollis*) is the sole crane species globally that inhabits, breeds, and flourishes in high plateau environments. There are many microbial communities in the gastrointestinal tract of birds, which play an important role in the health, nutrition, and physiology of birds. This study utilized high-throughput sequencing of the 16S rRNA gene to investigate and compare the core gut microbiota of black-necked cranes in two overwintering populations of Yunnan Province, China. A total of 34,297 operational taxonomic units were identified, belonging to 49 phyla, 130 classes, 276 orders, 365 families, and 775 genera. Alpha diversity analysis indicated significant differences in the diversity of gut microbial communities between the two wintering populations, with the central population exhibiting markedly higher diversity and richness compared to the eastern population. Principal coordinate analysis revealed a clear separation of the two populations of fecal samples, suggesting notable differences in microbial communities between the populations. The dominant phyla in the eastern population were Firmicutes, followed by Proteobacteria, whereas the central population was primarily dominated by Firmicutes, Proteobacteria, and Actinobacteria. At the genus level, *Lactobacillus* had the highest abundance in the intestinal microbiota of the two populations. Additionally, a variety of potential pathogenic bacteria was also found, including *Enterococcus*, *Acinetobacter*, *Campylobacter*, *Escherichia-Shigella*, and *Streptococcus*, which may pose a risk of transmission among local black-necked crane populations. Appropriate measures should be taken to protect the health of black-necked cranes and reduce the spread of diseases.

# INTRODUCTION

The black-necked crane (*Grus nigricollis*) is taxonomically classified within the order Gruiformes and belongs to the family Gruidae. It is designated as a first-class national protected species in China and listed as a vulnerable species on the IUCN Red List of Threatened Species. The black-necked crane is predominantly distributed within China, including Gansu Province (*Li-Xun et al., 2014*), Xinjiang Uygur Autonomous Region (*Han et al., 2017*), Tibet Autonomous Region (*Jia et al., 2019*), Qinghai Province (*Wei et al., 2021*), Yunnan Province (*Dong et al., 2016*), Sichuan Province (*Bai et al., 2022*) and Guizhou Province (*Gou et al., 2022*), with a minor population occurring in Bhutan (*Tshering & Jigme, 2014*). Its geographical distribution spans latitudes from 15°N to 37°N and longitudes from 86°E to 105°E (*Chen et al., 2023*). The primary breeding habitats of the black-necked crane are concentrated in the central and southwestern regions of Tibet, Qinghai, and select areas within Gansu Province (*Yang et al., 2023*). But it is the only alpine crane species that migrates for overwinterring on the Tibetan Plateau and the Yunnan-Guizhou Plateau (*Pu & Guo, 2023*). Based on their migratory patterns and wintering habitats, the black-necked crane is divided into three distinct wintering populations: the eastern, central, and western groups (*Pu & Guo, 2023*; *Kou et al., 2024*). In Yunnan province, the distribution of the overwintering black-necked cranes is characterized by two distinct populations. The Ruoergai Wetland in Sichuan serves as a crucial breeding ground for the eastern migratory population of black-necked cranes, this population migrates southward from the Ruoergai Wetland to the Dashanbao area in Yunnan Province for overwintering (*Pu & Guo, 2023*), the core area of which has a large area of swamp wetland. The main area is a subalpine grassland ecological type (*Zhao et al., 2021*), and the altitude is 3,000–3,200 m. The central population migrates southward from Longbaotan, Qinghai Province, along the Hengduan Mountains to Napahai (*Wang, Mi & Guo, 2020*). Napahai is a seasonal plateau swamp wetland. After the Napahai wetland enters winter, due to the reduction of water supply, the lake surface shrinks, and a large number of swamp meadows are exposed, with an average altitude of 3,260 m. The wintering area provides rich food resources and a safe habitat for black-necked cranes.

The gut microbiota often referred to metaphorically as the "second genome" of animals, underscores the complex and intimate relationship which shares with its host organism (*He et al., 2023*). The host organism creates a favorable environment that facilitates the survival and proliferation of its associated microbiota, which in turn plays a significant role in bolstering the host's immune defenses (*El Aidy, Dinan & Cryan, 2014*) and enhancing nutrient absorption (*Bäckhed et al., 2005*). The constituents and diversity of the gut microbiota are intricately correlated with the health status of the host. Dysbiosis within the gut microbiota can precipitate a detrimental imbalance, potentially heightening the host's vulnerability to a spectrum of diseases, such as conditions include obesity

(*Lozupone et al., 2012*; *Magne et al., 2020*), cancer (*Pevsner-Fischer et al., 2016*) and diabetes (*Forslund et al., 2015*). The composition and diversity of gut microbial communities are significantly influenced by a multitude of factors, such as dietary intake (*Liu, Gong & Li, 2020*; *Li et al., 2021*), the specific host species (*Hird et al., 2015*; *Gao et al., 2021*; *Wang et al., 2023a*), and the habitat (*Wu et al., 2018*). There are differences in the intestinal flora of black-necked cranes in different winter habitats (*Wang et al., 2020*). When the black-necked cranes in the same breeding area overwinter in different areas, the difference in the flora between the two overwintering areas is more obvious than that within the same area, indicating that external factors have a significant impact on the intestinal flora (*Kou et al., 2024*). During the overwintering period, the diversity of intestinal flora in the early overwintering period was higher than that in the late overwintering period; the ratio of Firmicutes to Bacteroides decreased, the abundance of Lactobacillus decreased, and the abundance of bacteria related to amino acid biosynthesis and acid metabolism increased (*Zhao et al., 2021*). In the meanwhile, the distinctive gastrointestinal architecture, intricate life cycles, and extensive migratory behaviors characteristic of birds are instrumental in shaping the high plasticity and reduced stability observed within their gut microbiota (*Hird, 2017*).

The conservation of the black-necked crane population has always been an important global concern. Research on the black-necked crane has primarily focused on aspects such as population dynamics, habitat utilization (*Kong et al., 2018*), dietary habits (*Dong et al., 2016*), and migratory patterns (*Pu & Guo, 2023*), while comparatively the studies on its gut microbiota is limited and insufficient. Although several studies have been conducted on the gut microbiota of black-necked cranes, the sample size remains insufficient, particularly regarding the collection of samples from various geographical regions. Additionally, there is a lack of research on the central population of Napahai black-necked cranes. Harnessing the gut microbiota as a strategic tool for the conservation of the black-necked crane necessitates, as a foundational step, the identification of the specific microbial communities inhabiting the gastrointestinal tract of this species. What's important, detecting the presence of pathogenic bacteria within the host is of paramount significant for the prevention and control of associated diseases. Yunnan Province, the primary habitat for black-necked cranes during their wintering season, is home to two distinct overwintering populations: the eastern and central regions. The objective of the present study is to employ the Illumina MiSeq platform for high-throughput sequencing of the V3–V4 region of the 16S rRNA gene, with the aim of analyzing and comparing the composition, diversity, and functional characteristics of the gut microbiota in two overwintering black-necked cranes populations located in Yunnan Province. The findings of this research will contribute to the identification of the core gut microbiota of black-necked cranes. Its purpose is to provide theoretical guidance for future studies on intestinal health, disease prevention, and the conservation of these cranes. By safeguarding the ecological environments of various wintering grounds, we can indirectly protect the microbial communities of black-necked cranes, thereby enhancing their health and increasing their survival rates.

## MATERIALS AND METHODS

### Study areas and sample collection

The study subjects were the black-necked cranes from the Dashanbao (DSB) area (27°18′–27°29′N, 103°14′–103°23′E, with an annual average temperature of 6.2 °C) in Zhaotong City and the Napahai (NPH) area (27°47′–27°55′N, 99°35′–99°40′E, with an annual average temperature of 6 °C) in Diqing Prefecture, Yunnan Province. Both regions exhibit plateau climatic characteristics and experience strong solar radiation. The sampling period was from February to March 2024.

The samples were collected from the nocturnal roosting areas and feeding grounds, prioritizing the selection of fresh fecal matter to reduce solar exposure. The central segment of each fecal sample was extracted and deposited into a sterile vial containing phosphate-buffered saline (PBS). After meticulous labeling, the samples were secured in a cooling box with dry ice for transportation to the laboratory, where they were archived at −80 °C for subsequent analysis.

### DNA extraction and species identification

All fecal samples collected from the black-necked cranes were subjected to microbial DNA extraction utilizing the ALFA-SEQ Magnetic Soil DNA Kit (Qiagen, Germantown, MD, USA), in accordance with the manufacturer's protocol. Polymerase chain reaction amplification was conducted on the DNA extracted from the fecal samples, targeting the Cytochrome b gene as identified in the literature (*Irwin, Kocher & Wilson, 1991*), to ascertain the species of the collected feces. A total of 40 fecal samples from black-necked cranes were collected, comprising 20 samples from DSB and 20 samples from NPH.

### 16S rRNA gene amplification and high-throughput sequencing

Using genomic DNA as a template, polymerase chain reaction amplification was conducted on the sample utilizing universal primers for the 16S rRNA V3–V4 region, specifically 338F (5′-ACTCCTACGGGAGGCAGCA-3′) and 806R (5′-GG ACTACHVGGGTWTCTAAT-3′). The PCR reaction system was composed of a total volume of 50 μL: premix taq (2×) 25 μL; forward primer 1 μL; reverse primer 1 μL; DNA 25 ng; nuclease free water added to 50 μL. Reaction procedure: 94 °C initial denaturation for 5 min; 30 cycles including (94 °C, 30 s; 53 °C, 30 s; 72 °C, 30 s); 72 °C, 8 min. The length and concentration of the PCR products were assessed using a 1.5% agarose gel. Subsequently, library construction was performed in accordance with the standard protocol of the ALFA-SEQ DNA Library Prep Kit. The prepared amplicon library was then sequenced utilizing the Illumina platform with paired-end sequencing (PE250).

### Data processing and bioinformatics analysis

The analysis of paired-end raw read data was conducted using fastp (version 0.14.1) for initial filtering. Subsequently, cutadapt (version 1.14) was employed to remove primers, resulting in the acquisition of paired-end clean reads following quality control measures. Non-compliant tags were filtered out utilizing usearch with the fastq_mergepairs function (version 10.0.240) to generate raw tags. A second round of sequence quality filtering on the

raw tags was performed using fastp (version 0.14.1) to yield clean tags. High-quality reads were then clustered into amplicon sequence variants (ASVs) with 100% sequence identity using DADA2 within the QIIME2 framework. The representative sequences of each ASV were subsequently compared against the SILVA (16S) database to obtain species annotation information based on the valid data. The sequences annotated as chloroplasts or mitochondria, along with those that cannot be classified at the phylum level, are excluded from the analysis. This procedure yields the final count of valid sequences and associated ASV classification data for each sample. Statistical analyses are conducted to evaluate shared and unique species, as well as to analyze community composition, utilizing R software. The alpha diversity of the microbial community is assessed through the Chao1 and Shannon indices. Furthermore, principal coordinate analysis (PCoA) is employed, applying both weighted UniFrac and unweighted UniFrac distance algorithms to examine the differences in microbial communities across various samples. The ANOSIM function is used to assess the significance of differences in community structure between groups. A significant difference analysis of species between two groups is performed using linear discriminant analysis effect size (LEfSe) analysis, with a linear discriminant analysis (LDA) score of 4 or greater and a $p$-value of less than 0.05, to identify species with significantly different abundances across groups. Subsequently, a Wilcoxon rank-sum test is conducted to evaluate the significance of the differences between the two groups. Functional analysis is carried out using the Kyoto Encyclopedia of Genes and Genomes (KEGG) by normalizing the ASV abundance table with PICRUSt2 to mitigate the influence of the copy number of the 16S marker gene within the species genome. Each ASV is then matched to the KEGG database to calculate the abundance of various functional categories. Field work was approved by the Yunnan Provincial Forestry and Grassland Bureau.

## RESULT

### Sequencing statistics

Utilizing the DADA2 plugin within QIIME2 for quality control, denoising, merging, and de-chimerization of all raw sequences from the fecal samples of the black-necked crane, a total of 4,749,410 high-quality sequences were obtained (Table 1). The average of 118,735 sequences per sample indicates that the study has successfully captured a significant amount of bacterial diversity.

### The composition of microbial communities in the feces of black-necked cranes

From a total of 40 fecal samples collected across two study areas, 34,297 ASVs were identified, representing 49 phyla, 130 classes, 276 orders, 365 families, and 775 genera. To investigate the microbial classification differences between the DSB and NPH populations, Venn diagrams revealed that 10,319 ASVs were obtained from the DSB population, while 24,531 ASVs were obtained from the NPH population across 40 samples. This analysis resulted in a total of 553 ASVs shared between the two populations (Fig. 1). There are significant differences in the relative microbial abundance of black-necked crane populations in DSB and NPH across various taxonomic levels (Fig. 2). At the phylum level,

**Table 1 Amplicon sequence variants and related sequence indexes in the fecal samples of crane.**

| Samples | Effective sequence number | Sequencing coverage/% | Q30/% |
|---|---|---|---|
| DSB | 121,601 | 99.84 | 95.17 |
| NPH | 115,869 | 99.68 | 95.29 |

the gut microbiota of black-necked cranes is predominantly composed of Firmicutes, Proteobacteria, Actinobacteria, and Fusobacteria, with Firmicutes being the most abundant phylum (Fig. 2A). The average relative abundance of Firmicutes in the DSB group was $76.57 \pm 13.14\%$, while in the NPH group it was $68.29 \pm 29.59\%$. At the order level, both groups exhibited higher numbers of Lactobacillales, Micrococcales, Enterobacteriales, and Fusobacteriales (Fig. 2B). At the family level, Lactobacillaceae showed the highest abundance, with the Enterococcaceae in the DSB group being more prevalent than in the NPH group (Fig. 2C). At the genus level, a total of 775 bacterial genera were identified from 40 sample sequences, The genus *Lactobacillus* exhibited relatively high abundance in both sample groups. The mean relative abundance of *Lactobacillus* in the DSB group was $42.01 \pm 24.06\%$, while that in the NPH group was $34.15 \pm 29.57\%$. The second most abundant genera were *Enterococcus* in the DSB group ($23.49 \pm 26.09\%$) and *Arthrobacter* in the NPH group ($6.03 \pm 17.75\%$) (Fig. 2D). The mean relative abundance of species composition at each taxonomic level for the two sample groups is provided in Table S1. Additionally, Wilcoxon rank-sum test was employed to analyze the significance of relative abundances exceeding 1% in the two groups of samples at the genus level, there were significant differences in the abundance of the genera *Bifidobacterium*, *Acinetobacter*, *Campylobacter*, *Enterococcus*, and *Stenotrophomonas* between the two regions.

## Gut microbiota diversity

The alpha diversity indices, specifically the Chao1 and Shannon indices, were calculated for each group to assess the diversity and richness of the microbial communities present in black-necked cranes from two distinct overwintering regions.

The results indicate a highly significant difference in both the diversity and richness of the gut microbial communities between the two groups of black-necked cranes(Chao1 index: $p = 0.001$, Shannon index: $p = 0.016$), with the NPH group exhibited significantly greater diversity and richness in their gut microbial communities compared to the DSB group (Figs. 3A, 3B). Furthermore, beta diversity analysis was conducted using the PCoA method to compare the composition of microbial communities across different samples. The analysis, based on unweighted and weighted UniFrac distances (Figs. 4A, 4B), revealed that fecal samples from the same geographical area tended to cluster together, thereby distinguishing the black-necked crane populations of Napahai and Dashanbao. ANOSIM was employed to evaluate the differences between the two groups, utilizing unweighted UniFrac distances ($R = 0.803$, $p = 0.001$) and weighted UniFrac distances ($R = 0.215$, $p = 0.001$). The results indicated significant differences in the composition of the intestinal microbial community among the various populations.

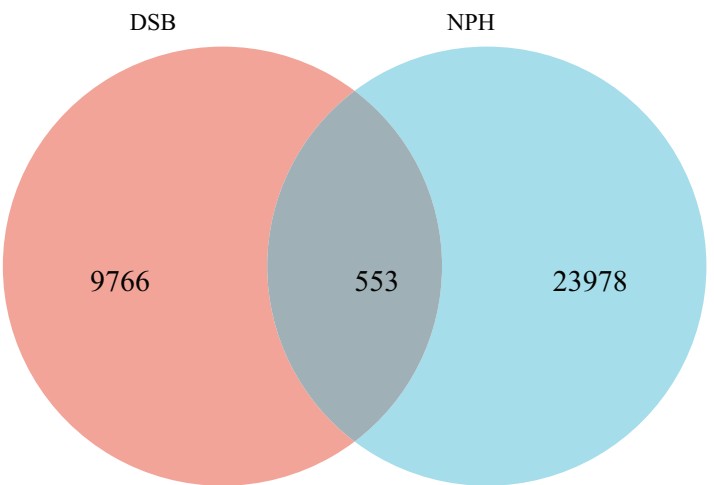

**Figure 1 Venn diagram of the bacterial community in the analyzed groups.** The number of shared or not shared amplicon sequence variants (ASVs; 100% sequence identity) is shown.

## Identification of potential microbial biomarkers

Multivariate statistical analysis, specifically the LEfSe analysis ($p < 0.05$; LDA score ≥ 4), was employed to identify various differentially abundant taxonomic biomarkers for the DSB (red) and NPH (green) groups (Figs. 5A, 5B). At the phylum level, the relative abundance of dominant phyla differed according to the overwintering site; the marker phylum for the DSB group was identified as Campilobacterota, while the NPH group was characterized by the presence of marker phyla including Actinobacteriota, Cyanobacteria, and Proteobacteria. At the genus level, the marker genera for the DSB group included *Enterococcus*, *Stenotrophomonas*, *Campylobacter*, and *Escherichia_Shigella*, whereas the NPH group was marked by the genera *Arthrobacter*, *Streptococcus*, *Bifidobacterium*, *Cetobacterium*, and *Psychrobacter*. These findings suggest that there are significant differences in the composition of microbial communities present in the feces of cranes from the two distinct regions.

## Predictive functional profiling of microbial communities

We conducted a predictive analysis of the gut microbiota functions of black-necked cranes from two distinct regions utilizing PICRUSt2. This analysis resulted in the annotation of six functional pathways at KEGG level 1 and twenty-eight functional pathways at KEGG level 2 (Fig. 6). The most abundant functional category identified was metabolism, followed by genetic information processing, cellular processes, environmental information processing, human diseases, and organismal systems. Notably, at KEGG level 2, significant differences were observed between the two sample groups in the areas of amino acid metabolism ($p = 0.01$), metabolism of terpenoids and polyketides ($p = 0.008$), and biosynthesis of other secondary metabolites ($p = 0.008$).

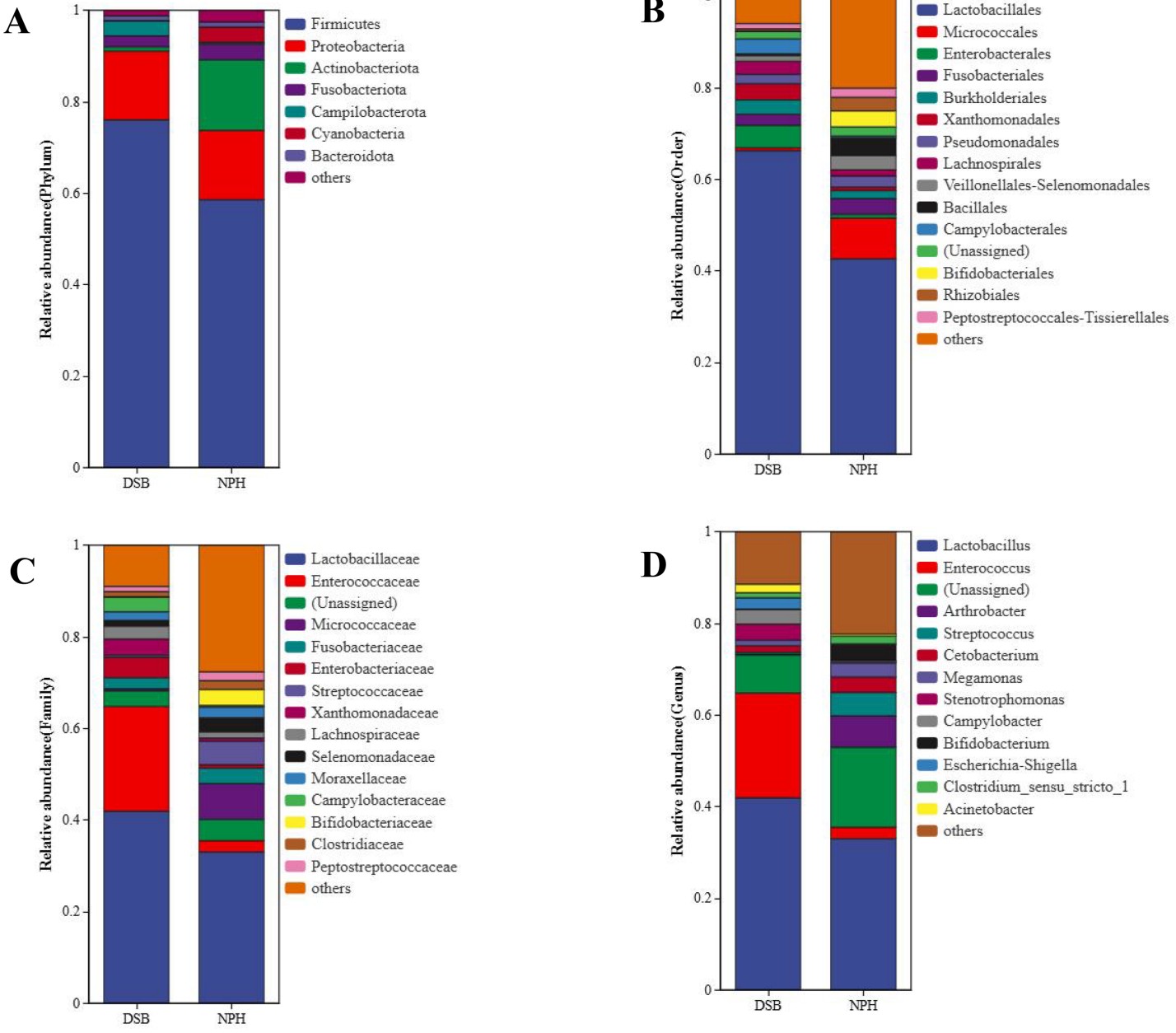

**Figure 2 Relative microbial abundance at different taxonomic levels among the two groups.** (A) Phylum; (B) order; (C) family; and (D) genus levels.

## DISCUSSION

The black-necked crane, revered as the "plateau fairy", serves as a flagship species for plateau wetland ecosystems. With an estimated global population of approximately 17,000 individuals, primarily within China, conservation efforts have been bolstered to enhance habitat quality and ecological restoration. Despite these measures, the species remains classified as Near Threatened (NT) by the International Union for Conservation of Nature

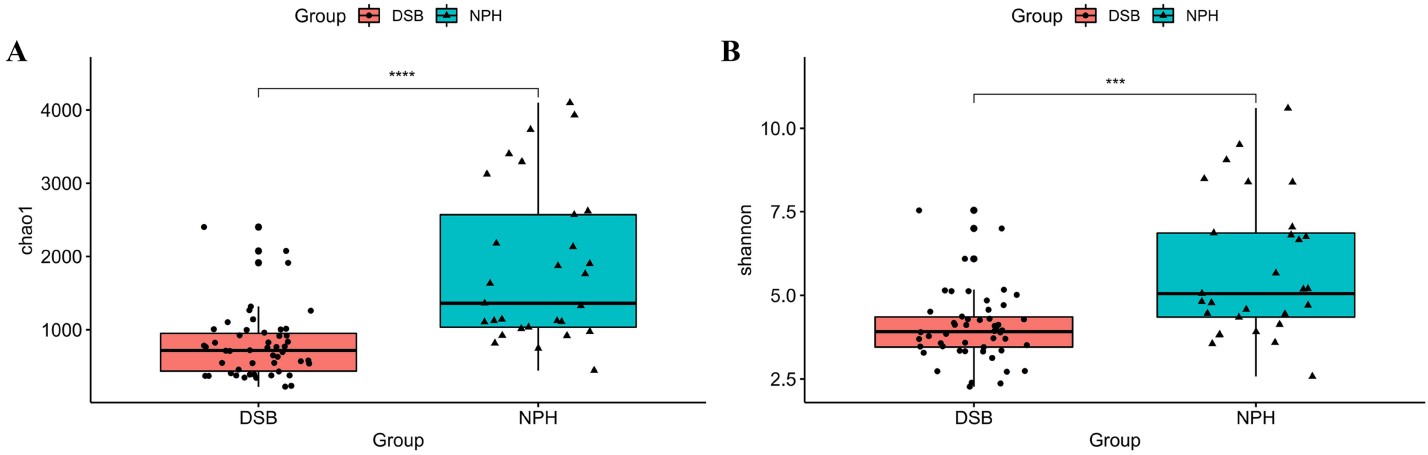

**Figure 3 Alpha diversity indices of gut microbial communities.** Chao1 index (A) and Shannon index (B) for DSB and NPH groups; ***$p < 0.001$, ****$p < 0.0001$.

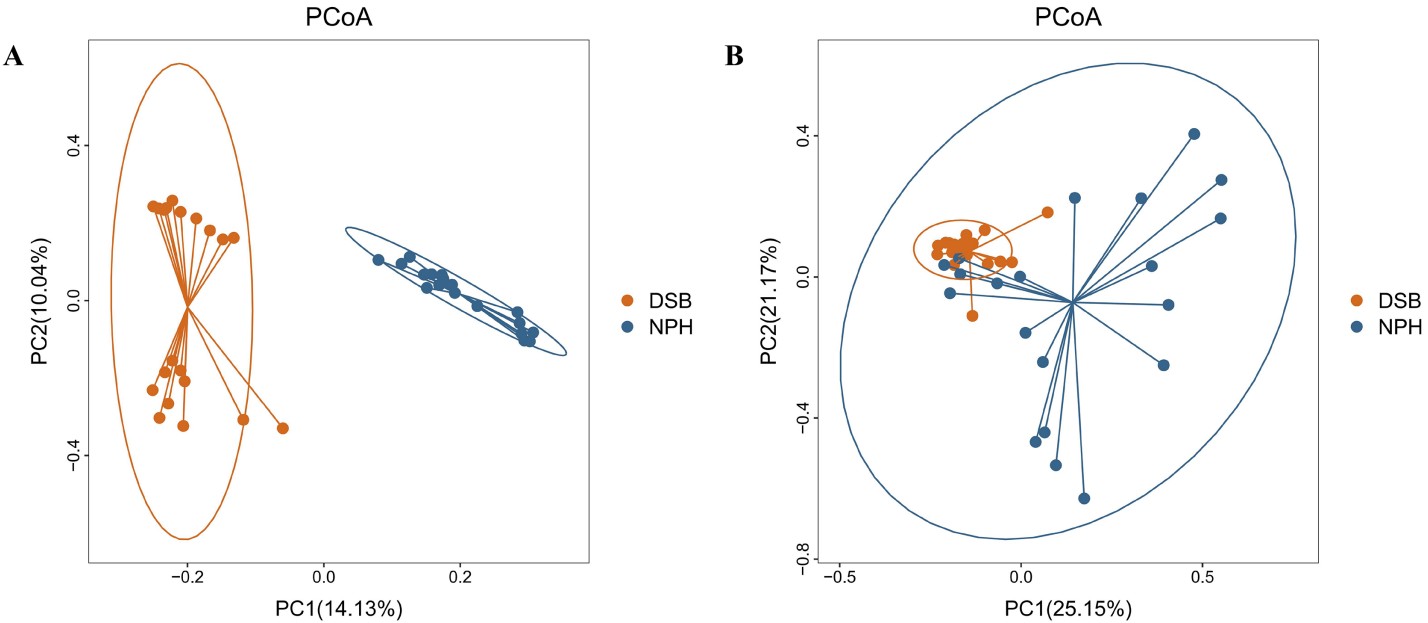

**Figure 4 PCoA plot based on two distance metric.** Unweighted (A) and weighted (B) UniFrac distance metric.

(IUCN), underscoring the necessity for ongoing conservation initiatives to safeguard population stability and growth. This study leverages the MiSeq sequencing platform to elucidate the gut microbiota of central and eastern black-necked cranes wintering in Yunnan Province, China, highlighting its significance in population protection, physiological ecology, and habitat assessment.

High-throughput 16S rRNA gene sequencing analysis enabled the exploration of gut microbial compositions in black-necked cranes without the need for cultivation. Based on this study, Firmicutes, Proteobacteria, Actinobacteriota and Fusobacteria constitute the
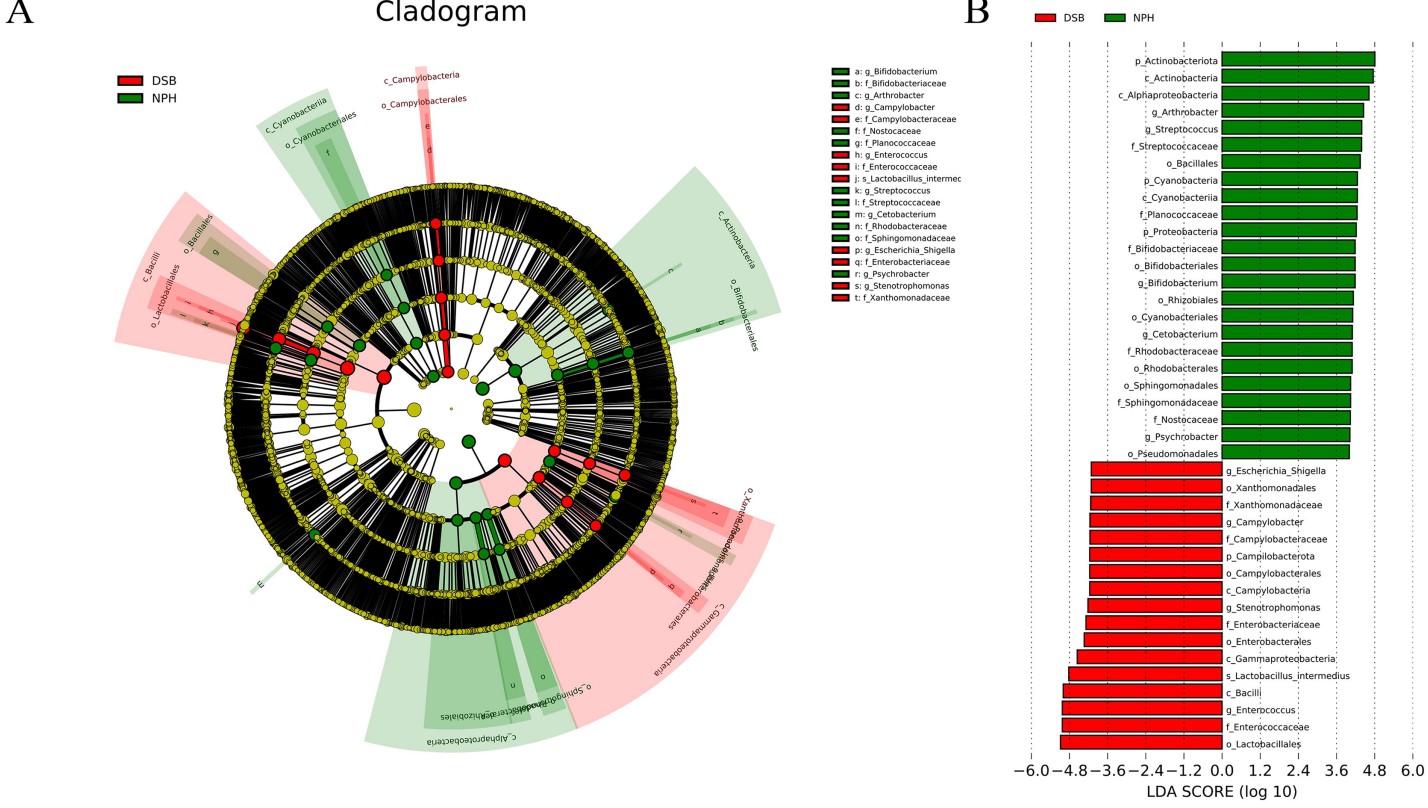

**Figure 5 Linear discriminant analysis (LDA) effect size (LEfSe) analysis.** (A) The microbial species had significant differences in the two sites. Different colors presents different groups, the species classification at the level of genus, family, order, class, and phylum were exhibited from the outside to the inside. (B) The plot from LEfSe analysis. The length of the bar column represents the LDA score. The figure exhibited the microbial with significant differences between the two breeding grounds (LDA score > 4.0).

core intestinal microflora of the black-necked crane population in Yunnan Province. This result was consistent with findings from previous studies involving various wild bird species, such as black-necked cranes in six regions of China (*Wang et al., 2020*), *Larus relictus* (*Liu et al., 2022*), *Grus virgo* and *Grus grus* (*Li et al., 2024*). This result indicates that these four phyla are present in the gut microbiota of various wild bird species, indicating their significance as functional bacteria in the gastrointestinal tract of birds. The abundance of Firmicutes in the eastern population is greater than that in the central population, Firmicutes play a crucial role in helping black-necked cranes digest and metabolize food that is rich in starch and fiber (*Ley et al., 2008*; *Turnbaugh et al., 2008*). This is related to the fact that the eastern population consumes a higher amount of carbohydrate-rich foods, enabling them to thrive in the harsh conditions of the plateau. Actinobacteriota were enriched in the central population. Studies have demonstrated that the increase in Actinobacteriota abundance is closely associated with obesity (*Turnbaugh et al., 2009*) and that they can produce short-chain fatty acids in the intestine by fermenting carbohydrates, such as dietary fiber. These fatty acids serve as the primary energy source for intestinal epithelial cells and help regulate the acid-base balance within the intestine, thereby enhancing the ecological environment for intestinal microorganisms

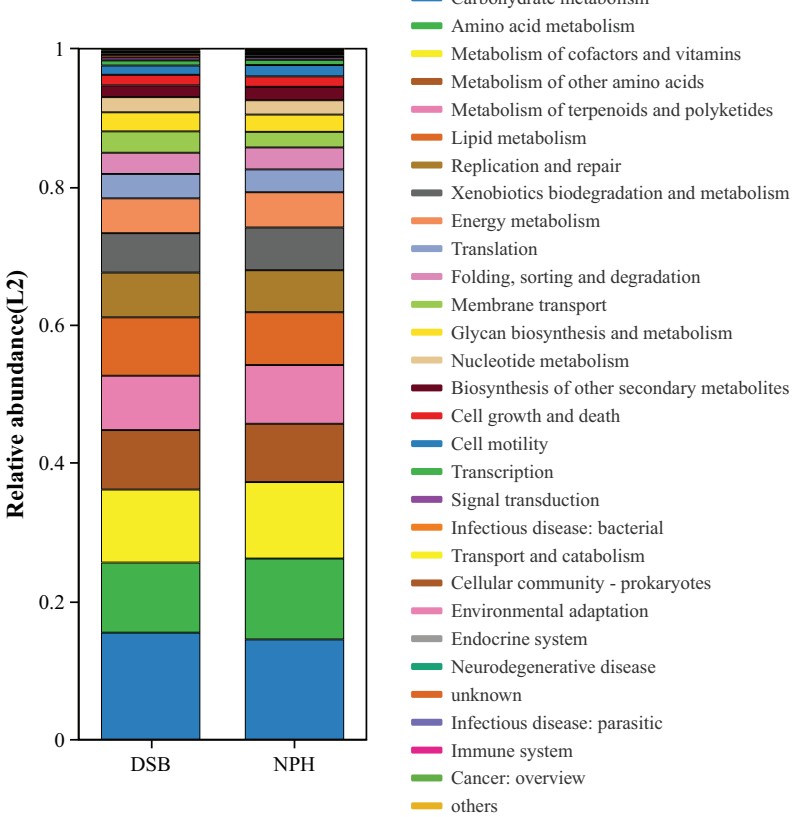

**Figure 6** Bacterial community functional prediction by Kyoto Encyclopedia of Genes and Genomes (KEGG) database.

(*Liu et al., 2021*). *Arthrobacter* is enriched in the intestinal flora of the central population, *Arthrobacter* of Actinobacteriota is widely distributed in soil, under cold environmental conditions, *Arthrobacter* species adapt to low temperatures through specific metabolic strategies. Current research indicates that these bacteria may manage transient nutrient deficiencies by accumulating energy storage compounds, such as glycogen (*Han et al., 2021*). The central population overwintering in the Napahai Wetland undergoes adaptive changes in their gut microbiota to cope with food shortages. The primary intestinal microflora of wintering black-necked cranes in Tibet includes Firmicutes, Proteobacteria, Fusobacteria, and unique Streptophyta (*Wang et al., 2023b*). The Streptophyta plays a crucial role as a primary producer in aquatic ecosystems. It serves as a direct or indirect food source for aquatic animals, including fish and insects, which are subsequently preyed upon by higher trophic level animals, such as birds. This dynamic may be linked to the ecosystem in which the western population of black-necked cranes overwinters. Additionally, various genera exhibit differing abundances across the two populations. PICRUSt2 functional predictions indicated that metabolism is the primary functional pathway of the core intestinal microbiota in black-necked cranes. While these functional predictions offer valuable insights, the existence and mechanisms of these metabolic

functions require further validation through metagenomic sequencing and metabolomic analysis.

The composition and diversity of avian gut microbiota are influenced by factors such as species, environment, diets, and regions of the digestive tract (*Sun et al., 2022*). In this study, PCoA results indicate variations in gut microbiota among individuals of the same species. Some studies have shown that changes in habitat during animal migration affect the gut microbiota of birds (*Price et al., 2017*; *Sun et al., 2022*), the influence of habitat environment on gut microbiota can sometimes surpass that of genetic factors (*Hird et al., 2014*). Black-necked cranes migrate to different regions for wintering, where they encounter diverse microorganisms influenced by their environmental conditions, including feeding habits, water sources, soil types, climate, and activities. The populations that overwinter in different areas are believed to have developed gut microbiota that are adapted to their specific environments. According to bird banding research, there are two migration routes for the black-necked crane that winters in Yunnan. The first route involves migrating southward from the Ruoergai Wetland in Sichuan to Dashanbao in northeastern Yunnan Province, further developing the eastern population. The second route entails migrating southward from Longbaotan in Qinghai along the Hengduan Mountains to Napahai in Diqing Prefecture, Yunnan Province, further developing the central population. Dietary factors are among the key elements that regulate the gut microbiota of black-necked cranes, which has been confirmed in other bird species (*Michel et al., 2018*; *Wu et al., 2023*). The black-necked crane primarily feeds on plants from the Solanaceae, Cyperaceae, Gramineae, Compositae, and Polygonaceae (*Wang et al., 2023a*). In the overwintering period, the reserve provides corn to the eastern population every day. In contrast, it only provides corn to the central population when heavy snow causes food shortages. The higher abundance of Firmicutes in the eastern population may be associated with the significant amount of corn in their diet (*Turnbaugh et al., 2008*; *Wu et al., 2023*). In the meanwhile, studies have shown that the composition of fecal microbiota is associated with altitude (*Zhang et al., 2016*). In our study, owing to the insignificant differences in altitude among the samples, it is challenging to accurately determine the impact of altitude on the gut microbiota composition of the black-necked cranes. Therefore, future studies should expand the sampling scope to cover a broader range of altitudes, thereby enabling a more comprehensive investigation into the potential effects of altitude on their gut microbial communities. The samples we collected are from the late overwintering period. Due to various factors, black-necked cranes have adapted to the environmental conditions of different overwintering locations, resulting in the formation of a unique intestinal microbial community. The two populations exhibit varying numbers of unique ASVs, which further supports the adaptation of the core intestinal microbial community of black-necked cranes to their overwintering environments.

There are significant differences in the richness and diversity of the microbial communities of black-necked cranes across various wintering areas. Alpha diversity analysis confirms that the diversity and richness of the gut microbiota in the central population are significantly higher than those in the eastern population. At the genus level,

*Lactobacillus* exhibits the highest abundance. *Lactobacillus* is a beneficial bacterium found in the intestine that possesses anti-inflammatory properties and protects the host from pathogens (*Bernardeau, Guguen & Vernoux, 2006*). Additionally, some potential pathogenic bacteria were found, such as *Enterococcus*, *Acinetobacter*, *Campylobacter*, *Escherichia-Shigella*, and *Streptococcus*, which can cause infections in various animals, including mammals and birds, leading to serious diseases (*Doughari et al., 2011*). *Streptococcus* is significantly enriched in the central population and can cause meningitis, bacterial pneumonia, endocarditis, and other infections (*Ge & Sun, 2014*). *Escherichia-Shigella*, *Campylobacter*, and *Enterococcus* are significantly enriched in the eastern population. These pathogens can cause sepsis, urinary tract infections, diarrhea, and other illnesses (*Arias & Murray, 2012*; *Li et al., 2025*). Further research is needed to determine whether these potential pathogens can cause physiological diseases in black-necked cranes. Research has identified a negative correlation between the diversity of gut microbiota and the presence of pathogenic bacteria in the gut (*Xiang et al., 2019*). The intestines of healthy hosts harbor diverse bacterial communities, establish a state of equilibrium, and diseases can disrupt this balance, leading to a reduction in gut bacterial diversity (*Mangin et al., 2004*). Studies have found that these pathogenic bacteria can spread among wild birds, free-range poultry, livestock, and humans (*Kobuszewska & Wysok, 2024*). Additionally, bird migration behaviors facilitate the transmission of pathogens across different geographical regions (*Qiu et al., 2024*). Therefore, regular monitoring of the gut microbiota of black-necked cranes is vital to prevent the emergence and spread of pathogenic microorganisms.

In response to the conservation needs of the black-necked crane, China has implemented a variety of strategies aimed at protecting both the population and its habitat. These strategies include the establishment of nature reserves, rescue operations, and the provision of compensation for ecological services provided by wetlands. Due to the enhancement of protective measures, the population of black-necked cranes is currently on the rise. Suggestions for the long-term monitoring of the black-necked crane population include the protection of water bodies in wintering areas, prevention of pollution from agricultural and industrial activities, monitoring the impact of climate change on the population, and developing adaptation strategies to address climate change. For the eastern population, it is essential to raise awareness among residents in protected areas regarding the conservation of black-necked cranes and to minimize human interference in their overnight roosting and foraging areas. For the central population, it is crucial to ensure a sufficient food supply for black-necked cranes during the wintering period. In the conservation of the black-necked crane and the evaluation of its population health, it is essential to monitor alterations in the gut microbiota and to implement strategies aimed at minimizing pathogenic microorganisms within the environment.

## CONCLUSIONS

In summary, this study employed high-throughput sequencing analysis on fecal samples collected from black-necked cranes wintering in the eastern and central population of Yunnan Province. This study found that the gut microbiota is primarily composed of

Firmicutes, Proteobacteria, Actinobacterota, and Fusobacteria. There are core and unique microbial communities in the intestines of different populations of black-necked cranes. The findings revealed significant differences in gut microbiota among the various wintering populations, with multiple factors contributing to these variations, the factors include diet, habitat, and environmental conditions. A variety of potential pathogens were identified in the two populations. These results will establish a foundation for studying the role of intestinal microorganisms in black-necked cranes and provide a theoretical basis for their conservation, as well as for the prevention and control of related diseases.

## ACKNOWLEDGEMENTS

Thank you to the dedicated staff at each black-necked crane sampling point for their hard work.

### Funding

This work was funded by the National Natural Science Foundation of China (82460392); The Youth Talent Program of Yunnan "Ten-thousand Talents Program" (YNWR-QNBJ-2020–089). The funders had no role in study design, data collection and analysis, decision to publish, or preparation of the manuscript.

### Grant Disclosures

The following grant information was disclosed by the authors:
National Natural Science Foundation of China: 82460392.
The Youth Talent Program of Yunnan "Ten-thousand Talents Program":
YNWR-QNBJ-2020–089.

### Competing Interests

The authors declare that they have no competing interests.

### Author Contributions

- Ruimei Wang conceived and designed the experiments, performed the experiments, prepared figures and/or tables, authored or reviewed drafts of the article, and approved the final draft.
- Yixuan Wang conceived and designed the experiments, analyzed the data, authored or reviewed drafts of the article, and approved the final draft.
- Lulu Deng performed the experiments, prepared figures and/or tables, and approved the final draft.
- Binghui Wang conceived and designed the experiments, prepared figures and/or tables, authored or reviewed drafts of the article, and approved the final draft.
- Mingfei Shi analyzed the data, authored or reviewed drafts of the article, and approved the final draft.
- Zeya Yang analyzed the data, authored or reviewed drafts of the article, and approved the final draft.

- Dong Hu analyzed the data, authored or reviewed drafts of the article, sample collection, and approved the final draft.
- Zijiao Zhao performed the experiments, authored or reviewed drafts of the article, sample collection, and approved the final draft.
- Ruiling Yuan performed the experiments, prepared figures and/or tables, authored or reviewed drafts of the article, and approved the final draft.
- Jiuxuan Zhou conceived and designed the experiments, prepared figures and/or tables, authored or reviewed drafts of the article, and approved the final draft.

### Field Study Permissions

The following information was supplied relating to field study approvals (*i.e.*, approving body and any reference numbers):

Field work was approved by the Yunnan Provincial Forestry and Grassland Bureau.

### Data Availability

The sequence data is available at GenBank: PRJNA1212951.

### Supplemental Information

Supplemental information for this article can be found online at http://dx.doi.org/10.7717/peerj.19520#supplemental-information.

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
