# Peer review of "Comparative analysis of fecal microbiota of central and eastern black-necked cranes (*Grus nigricollis*) wintering in Yunnan Province, China"

_PeerJ, doi:10.7717/peerj.19520_

## Round 0.1 · original submission · Major Revisions

The reviewers all agree that the study is of interest and has merit. However, each has several concerns with the manuscripts that will require addressing.

Reviewer 1 ·

Basic reporting

The manuscript conducted high-throughput sequencing of the V3-V4 region of the 16S rRNA gene in feces samples collected from black-necked cranes wintering at Dashanbao and Napahai in Yunnan, which are situated along two different migration routes. It analyzed the differences in the composition of intestinal microbial communities between the two distinct populations of wintering black-necked cranes. The research possesses a clear methodology and logical framework, and the findings are of great significance for understanding the role of intestinal microbiota in the environmental adaptation of black-necked cranes. However, some parts of the manuscript exhibit unclear logical relationships, an insufficiently in-depth description of the background in the introduction, and a lack of thorough and rigorous discussion. Specifically, the issues are as follows:
Abstract
Due to the lack of supporting evidence in the analysis of some results during the discussion, the conclusions presented in the abstract lack credibility. The authors are required to further refine the conclusions based on the modifications made to the results and discussion sections.
Introduction
The introduction lacks depth and does not adequately summarize the current research status on the intestinal microbiota of black-necked cranes. This study primarily focuses on comparing the intestinal microbial diversity of black-necked cranes at their main wintering grounds along two different migration routes. However, the background information on black-necked cranes at these two study sites is not clearly presented in the introduction. Please further enhance the introduction by providing more comprehensive background information.
Materials and Methods
Lines 165: Which mentions using PICRUSt for functional prediction, as the results section indicates the use of PICRUSt2 instead.
There is a lack of significance testing in the PCoA (Principal Coordinate Analysis) analysis.
Results
Lines 183-192: When presenting the richness of bacterial communities, it is advisable to express the data using the mean ± standard deviation. This allows for a clearer understanding of the variability and central tendency of the data.
Lines 192-194: What statistical method for testing differences was used to demonstrate significant differences in these bacterial communities between the two different wintering populations? Please clarify.
Lines 208-212: Supplement the difference analysis by conducting significance testing on the PCoA (Principal Coordinate Analysis) results.
Lines 214-225: LEfSe (Linear Discriminant Analysis Effect Size) is primarily used to identify biomarkers that are abundant in different groups and have a significant impact on the overall composition. Here, there are some important biomarkers present in the intestinal microbiota of the two wintering populations at various taxonomic levels, which should be accurately described. Additionally, the discussion should analyze the reasons for the significantly higher abundance of these biomarkers in different groups.
Lines 235-237: Such an interpretation lacks rigor and is unsupported by evidence.
Discussion
Lines 248-283: there is an inadequate explanation of the relationship between the bacterial communities with significantly higher abundance in different populations and the habitat of the host itself. Instead, it merely describes the functions of these bacterial communities without explaining the issues reflected by the differences in abundance between different groups.
Lines 277-283: The statement is unclear and not well-supported. Simply stating that differences in gut microbiota abundance at the genus level and a higher enrichment rate of metabolic pathways in PICRUSt2 functional prediction results can help the host accumulate energy to cope with the upcoming migration lacks evidence and rigor.
Lines 299-303: Since they belong to different wintering grounds, there are still significant differences in the wintering environments. Additionally, according to the author's description in the methods, the sampling time should be in the middle to late stages of wintering. From this perspective, the differences in the number of unique ASVs are more likely to be related to adaptation to the wintering environment rather than solely attributed to their belonging to different migration routes.
Lines 303-312: The explanation is too confusing, and many descriptions lack evidence. Since there is a lack of research or cited data on foraging strategies in this study, it lacks a basis to directly state that they have consumed a large amount of fish. The authors should carefully review and revise their arguments.
Lines 312-315: In this study, a comparative analysis of the gut microbiota of black-necked cranes at different altitudes was not directly conducted, and I am unclear how the author has determined that there is no correlation with altitude changes. Please explain.
Lines 320: How is 'dominant genera' defined?
Lines 346-350: Lacks evidence.
Lines 366-372: Although research on the gut microbiota of wild birds is not very in-depth, there are still reports on the composition and function of the gut microbiota of wintering migratory birds, including black-necked cranes. Please have the authors carefully review the current research status.
Conclusions
The conclusion lacks evidence and needs to be further refined based on the modifications made in the discussion section.
other
The resolution of the figures is too low, causing the key information presented in the text to be unclear. Further adjustments and enhancements to the figure resolution are required.

Experimental design

NO

Validity of the findings

NO

Additional comments

NO

Reviewer 2 ·

Basic reporting

This study has certain academic value, as it fills some research gaps in this field by comparing the gut microbiota communities of black-necked crane populations in two overwintering sites in Yunnan using 16S rRNA sequencing technology. The research methods are relatively rigorous and comprehensive, including Chao1/Shannon indices, PCoA, and LEfSe, and the quality of the raw data meets the high-throughput standards. In terms of innovation, it is the first to compare the populations in central Yunnan (NPH) and eastern Yunnan (DSB), emphasizing the impact of the environment on the core microbiota and pathogen prevalence. Moreover, the functional prediction carried out by PICRUSt links the microbial composition to energy metabolism, thereby enhancing the ecological interpretation of the research results. However, there are still some shortcomings in this paper, which are listed here for discussion with the author.

Experimental design

1. About the Data Collection
The study collected 20 fecal samples per wintering site (total n=40) but failed to provide statistical justification for this sample size (e.g., power analysis or rarefaction curves to confirm sufficient sampling depth). Small sample sizes may lead to Type II errors (failure to detect true microbial diversity differences between groups).
All samples were collected within a single wintering season (February–March 2024). These risks confounding effects from short-term environmental fluctuations (e.g., atypical weather or temporary food provisioning), potentially undermining claims about "stable" microbial adaptations.
2. About the Study Design
The use of 338F/806R primers for amplifying the V3–V4 region was not validated for avian gut microbiota. Primer mismatches with anaerobic taxa (common in bird intestines) could skew taxonomic profiles.
PICRUSt2 relies on reference genomes (e.g., KEGG) that poorly represent avian gut microbiomes. The authors did not validate prediction accuracy (e.g., via metagenomic sequencing or comparisons to host metabolic data), raising concerns about inferred "energy metabolism" functions.
No data on environmental microbes (e.g., water/soil microbiota), dietary composition (e.g., δ¹³C stable isotopes), or anthropogenic pressures (e.g., tourist density, feed types). This precludes disentangling host-intrinsic vs. environment-driven microbial differences.

Validity of the findings

1. About the Conclusions
Claims about "higher pathogenic risk" in eastern populations (DSB) lack direct evidence:
Pathogenicity was inferred solely from taxonomic abundance (e.g., Enterococcus), without culturing, virulence gene screening, or clinical health data from cranes.
No comparative analysis with sympatric livestock/wildlife to confirm human activity as the source of pathogens.
The assertion that microbiota "support high-altitude adaptation via metabolic functions" is speculative: no explicit KEGG pathways (e.g., short-chain fatty acid biosynthesis) or host physiological measurements (e.g., metabolic rates) directly link microbial functions to adaptation. PICRUSt2 predictions remain hypothetical without experimental validation.

Additional comments

The study’s core findings are valuable but currently limited by insufficient environmental data, unvalidated functional inferences, and unsupported causal claims. Revisions should prioritize methodological transparency, expanded data collection, and tempered conclusions to meet PeerJ’s standards.

·

Basic reporting

no comment

Experimental design

no comment

Validity of the findings

no comment

Additional comments

The researchers collected fecal samples from two overwintering black-necked crane populations in central and western Yunnan, China, and conducted a comparative analysis of their gut microbial communities using 16S rRNA gene amplicon sequencing (targeting the V3-V4 regions). This study can enhance our understanding of the gut microbiota in wild migratory birds and provides a foundation for further research on black-necked cranes.
1.The abstract states that certain genera (e.g., Enterococcus) are "highly abundant pathogenic bacteria." However, the main text does not provide specific descriptions of their relative abundances. The criteria for defining "highly abundant" need to be clarified.
2.In line 172, the article only reported the average number of sequencing reads per sample but failed to evaluate the adequacy of sequencing depth using rarefaction curves or species accumulation curves. This omission makes it impossible to assess whether the sequencing data sufficiently captured the microbial diversity within the samples. If the curves do not reach a plateau, it would indicate insufficient sequencing depth, potentially leading to the omission of low-abundance taxa and compromising the reliability of diversity analyses (e.g., Chao1 and Shannon indices). Furthermore, if there are substantial differences in sequencing depth between samples (even if average values are similar), direct comparisons of alpha diversity may introduce bias.
3. in line 185-186, the description of "the average relative abundances" is ambiguous. It is unclear whether these values refer to the combined abundance of the four mentioned phyla (Firmicutes, Proteobacteria, Actinobacteria, Fusobacteria), or the abundance of the dominant phylum Firmicutes alone.
4.The manuscript contains inconsistent formatting of the statistical term "p-value." For instance, line 205 uses a lowercase "p" (e.g., "p=0.001"), while other sections may use uppercase "P". To align with academic conventions and enhance clarity, please standardize the formatting throughout the text.
5. In line 208, the study demonstrates differences in gut microbial communities between the two black-necked crane populations using Principal Coordinate Analysis (PCoA). However, it does not employ statistical tests (e.g., PERMANOVA or ANOSIM) to quantify the explanatory power of the observed differences between the two groups. PCoA can visualize sample distances, but it cannot validate the statistical significance of group-level differences or calculate the contribution of grouping variables to community variation.
6. In line 248, the phrase "High-throughput 16S rRNA sequencing" should be "High-throughput 16S rRNA gene sequencing" for technical accuracy.
7.In line 282, the authors conclude that "black-necked cranes require substantial energy reserves to prepare for migration" based on functional prediction result that metabolism constitutes the largest proportion of microbial activity. However, the study does not analyze dynamic changes in metabolic functions across different overwintering stages (e.g., pre-migration, mid-migration, and post-migration phases). Such stage-specific variations in energy demands are critical to the adaptive strategies of migratory species. The lack of empirical data on temporal metabolic shifts weakens the conclusiveness of this claim, rendering it overly speculative. To strengthen the argument, it is recommended to cite relevant literature to substantiate this hypothesis.
8.The font size in Figure 5A is too small, making it difficult to read the labels.

---

## Round 0.2 · Minor Revisions

The reviewers were pleased that the majority of their previous concerns had been addressed in this revision, but still had a couple of points that required addressing.

On the question of altitude, and its correlation with differences in the gut microbiota, I can understand the viewpoints of both the authors and the reviewer. Would it be acceptable to both parties to state that given the lack of significant difference in altitude between samples that it is unlikely that altitude has contributed to changes in fecal microbiota composition, but to answer this question conclusively, further work would be required with a dataset designed specifically for this analysis?

Reviewer 1 ·

Basic reporting

After the revisions, the author has addressed most of the queries, but there are still some issues that remain inadequately resolved and require further refinement by the author.
1、 Lines 54-61: The statement lacks referential basis. Additionally, it should be noted that Ruoergai in Sichuan is also an important breeding ground for the eastern migratory population of Black-necked Cranes.
2、 Lines 216-220: The author lacks the examination of pathogenicity indicators for these bacterial in the results section. In the discussion, the author judges these bacterial as potentially pathogenic based on studies in mammals or other birds. Therefore, in the results section, the author should focus more on the relative abundance of these bacterial rather than directly stating their potential pathogenicity.
3、 Lines 335-343: I still don't understand the author assertion that there is no correlation with altitude. The author explains that since there is no difference in altitude between the Black-necked Crane habitats in Dashanbao and Napahai, but there are significant differences in their gut microbiota, they speculate that there is no correlation between gut microbiota and altitude. I believe this speculation is unreliable. Since the author did not conduct a comparative analysis across different altitude ranges, it is not possible to draw a conclusion about the correlation with altitude.

Experimental design

no

Validity of the findings

no

Additional comments

no

---

## Round 0.3 · accepted · Accept

The reviewer was happy that all concerns that they had raised had been addressed by the authors. This manuscript is now ready for publication.

Reviewer 1 ·

Basic reporting

The author has already made serious revisions and improvements, with no further comments.

Experimental design

no

Validity of the findings

no

Additional comments

no